# Robots That Ask For Help: Uncertainty Alignment for Large Language Model Planners

**Allen Z. Ren**[1,2], **Anushri Dixit**[1], **Alexandra Bodrova**[1], **Sumeet Singh**[2], **Stephen Tu**[2],
**Noah Brown**[2], **Peng Xu**[2], **Leila Takayama**[2], **Fei Xia**[2], **Jake Varley**[2], **Zhenjia Xu**[2],
**Dorsa Sadigh**[2], **Andy Zeng**[2], **Anirudha Majumdar**[1,2]
[1]Princeton University, [2]Google DeepMind

**Abstract:** Large language models (LLMs) exhibit a wide range of promising capabilities — from step-by-step planning to commonsense reasoning — that may provide utility for robots, but remain prone to *confidently* hallucinated predictions. In this work, we present KNOWNO, a framework for measuring and aligning the uncertainty of LLM-based planners, such that they know when they don't know, and ask for help when needed. KNOWNO builds on the theory of conformal prediction to provide statistical guarantees on task completion while minimizing human help in complex multi-step planning settings. Experiments across a variety of simulated and real robot setups that involve tasks with different modes of ambiguity (e.g., from spatial to numeric uncertainties, from human preferences to Winograd schemas) show that KNOWNO performs favorably over modern baselines (which may involve ensembles or extensive prompt tuning) in terms of improving efficiency and autonomy, while providing formal assurances. KNOWNO can be used with LLMs out-of-the-box without model-finetuning, and suggests a promising lightweight approach to modeling uncertainty that can complement and scale with the growing capabilities of foundation models.[1]

**Keywords:** Language-based planning, uncertainty estimation, conformal prediction

## 1 Introduction

How can we endow our robots with the ability to *know when they don't know*? Accurately modeling and accounting for uncertainty is a longstanding challenge towards robots that operate reliably in unstructured and novel environments. In this work, we study this challenge in the context of *language-instructed* robots. Language provides a natural and flexible interface for humans to specify tasks, contextual information, and intentions, while also allowing us to provide help and clarification to robots when they are uncertain.

Recently, approaches that leverage large language models (LLMs) for planning [1, 2] have demonstrated the ability to respond to natural and unstructured language instructions to generate temporally extended plans. These approaches enable leveraging the vast amount of prior knowledge and rich context embedded in pretrained LLMs, and lead to substantial abstract reasoning capabilities. However, one of the major challenges with current LLMs is their tendency to *hallucinate*, i.e., to *confidently* generate outputs that are plausible but incorrect and untethered from reality. Such false confidence in incorrect outputs poses a significant challenge to LLM-based planning in robotics. Natural language instructions in real-world environments often contain a high degree of inherent ambiguity, and confidently following an incorrectly constructed plan could lead to undesirable or even unsafe actions.

As an example, a robot tasked with heating food may choose to place a metal bowl in the microwave, potentially damaging the microwave or even causing a fire (Fig. 1). Instead of following such a plan, we would like our robots to know when they don't know and ask for clarification instead (e.g., ask which bowl is safe to be placed in the microwave). Prior work in language-based planning either does not seek such clarifications [1] or does so via extensive prompting [2], which requires careful prompt engineering to prevent the robot from excessively relying on seeking assistance. Moreover, prior approaches do not provide a way to ensure that asking for help results in a desired level of task success. We formalize

---

[1]Webpage with additional results and videos: https://robot-help.github.io

7th Conference on Robot Learning (CoRL 2023), Atlanta, USA.

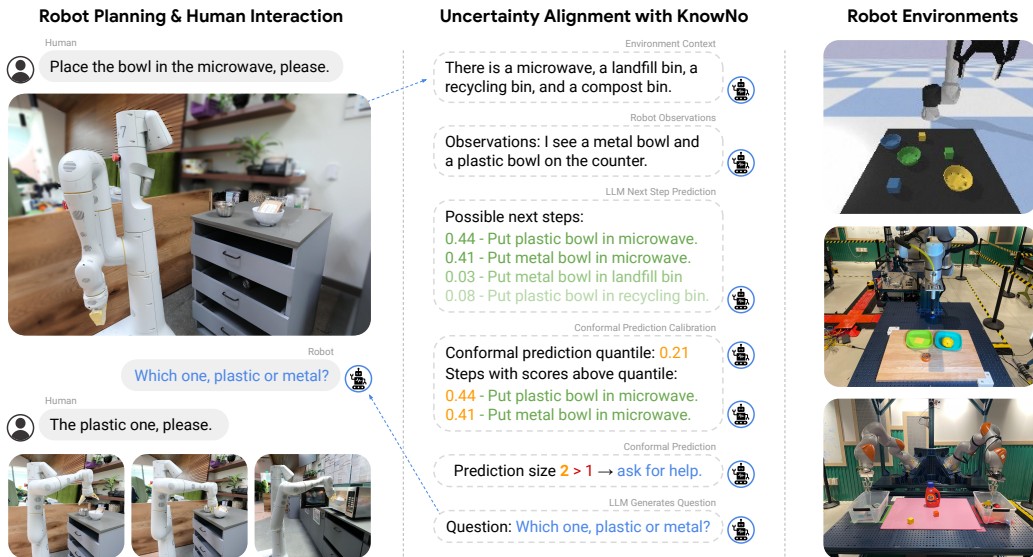

**Robot Planning & Human Interaction**

Human
Place the bowl in the microwave, please.

Robot
Which one, plastic or metal?

Human
The plastic one, please.

**Uncertainty Alignment with KnowNo**

Environment Context
There is a microwave, a landfill bin, a recycling bin, and a compost bin.

Robot Observations
Observations: I see a metal bowl and a plastic bowl on the counter.

LLM Next Step Prediction
Possible next steps:
0.44 - Put plastic bowl in microwave.
0.41 - Put metal bowl in microwave.
0.03 - Put metal bowl in landfill bin.
0.08 - Put plastic bowl in recycling bin.

Conformal Prediction Calibration
Conformal prediction quantile: 0.21
Steps with scores above quantile:
0.44 - Put plastic bowl in microwave.
0.41 - Put metal bowl in microwave.

Conformal Prediction
Prediction size 2 > 1 → ask for help.

LLM Generates Question
Question: Which one, plastic or metal?

**Robot Environments**

Figure 1: KNOWNO uses Conformal Prediction (CP) to align the uncertainty of LLM planners. Given language instructions, an LLM generates a set of possible next steps (with scores above a certain CP calibration quantile). If there are more than one in the set, the robot asks for help, then readjusts accordingly. Experiments show that KNOWNO works well with multiple embodiments, across a variety of ambiguous situations, in both simulated and real setups.

these challenges via two desiderata: (i) *calibrated confidence:* the robot should seek sufficient help to ensure a statistically guaranteed level of task success specified by the user, and (ii) *minimal help:* the robot should minimize the overall amount of help it seeks by narrowing down possible ambiguities in a task. We collectively refer to these sufficiency and minimality conditions as *uncertainty alignment*.

**Statement of contributions.** We propose KNOWNO— *Know When You Don't Know* — a framework for aligning the uncertainty of LLM-based planners utilizing the theory of *conformal prediction (CP)* [3, 4]. We make the following contributions: **(1)** Given a language instruction, we utilize a pre-trained LLM with uncalibrated confidence to generate a *set* of possible actions for the robot to execute next. We demonstrate how to use CP to select a subset of these options, which allows the robot to decide an action to execute (if the subset is a singleton) or to ask for help otherwise. **(2)** We prove theoretical guarantees on calibrated confidence in both single-step and multi-step planning problems: with a user-defined confidence $1-\epsilon$ over instructions, the robot performs all actions for a given instruction correctly by asking for help when necessary. CP also minimizes the average size of prediction sets, thus addressing the goal of minimal help. **(3)** We evaluate KNOWNO in both simulation and hardware with a suite of language-instructed manipulation tasks with various types of potential ambiguities (e.g., based on spatial locations, numerical values, attributes of objects, and Winograd schemas). Experiments across multiple settings and embodiments validate the ability of KNOWNO to provide statistically guaranteed levels of task success while significantly reducing the amount of help required to achieve a given success level by $10-15\%$ as compared to baseline approaches.

## 2 Overview: Robots that Ask for Help

**Language-based planners.** Language model planners can generate step-by-step robot plans, where each step $y$ is composed of variable-length sequences of symbols $(\sigma_1, \sigma_2, ..., \sigma_k)$, e.g., text tokens as input to a language-conditioned policy [1] (see Fig. 1), or robot code executed by an interpreter [5]. Pretrained autoregressive LLMs predict each step $y$, whose joint probability over tokens can be factorized as the product of conditional probabilities of next token prediction $p(y) = \prod_{i=1}^{k} p(\sigma_i \mid \sigma_1, ..., \sigma_{i-1})$. Here, we are interested in characterizing the uncertainty of next step prediction $p(y)$. The distribution of $p(y)$ values (i.e., perplexity) remains highly sensitive to variable-length $k$; hence $p(y)$ on its own serves as a rather poor scoring function [6] particularly when steps in a plan are expressed in natural language (our experiments in Section A8 also show that using $p(y)$ directly for calibration leads to poor performance).

**Planning as multiple-choice Q&A.** We can address this length bias with a simple trick, whereby we can prompt the LLM to generate (either with high-temperature sampling, or with another LLM instance)

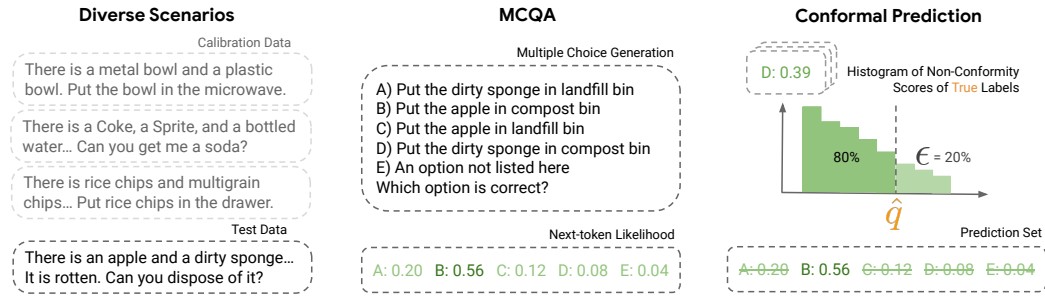

Figure 2: KNOWNO formulates LLM planning as MCQA by first prompting LLM to generate plausible options, then asking it to predict the correct one. Using the next-token likelihoods collected with a calibration dataset, CP provides the quantile value $\hat{q}$ such that all options with a score $\geq 1-\hat{q}$ are included in the prediction set in a test scenario.

a *set* $\{y^i\}$ of candidate next steps (e.g., "Put plastic bowl in microwave", "Put metal bowl in microwave", etc., in Fig. 1), and then format the task of choosing among them as multiple-choice Q&A (MCQA). This eliminates plans that the LLM considers unlikely and reduces the problem of next-step prediction down to a single next-token prediction — aligning with LLM log-likelihood loss functions and LLM training data (e.g., MCQA datasets [7, 8]). These probabilities can serve as normalized scores that can be used by various uncertainty quantification methods such as thresholding, ensemble methods, and others. In this work, we use these normalized scores within a conformal prediction (CP) framework. Specifically, CP uses a held-out calibration set of example plans in different scenarios to generate a reduced prediction set of plans among $\{y^i\}$ (Fig. 2). The LLM is certain if this prediction set is a singleton, and triggers help from a human otherwise.

**Robots that ask for help.** In this work, we show that LLM planning — combined with CP for uncertainty estimation — can effectively enable robots to interact with an environment, and ask for help when needed. The environment $e$ can be formulated as a partially observable Markov decision process (POMDP): at any given state $s_t$ at time $t$, given a user instruction $\ell$, the robot executes an action $a_t$ according to a policy $\pi$, then transitions to a new state $s_{t+1}$. Our policy $\pi$ is composed of four parts (Fig. 1):

1. Multiple-choice generation: An LLM generates a diverse set of candidate plans labeled with 'A', 'B', 'C', 'D', and an additional possible plan, 'E) an option not listed here', which is appended post-hoc. We denote the set of labels by $\mathcal{Y} := \{$'A','B','C','D','E'$\}$. These plans are generated by prompting the LLM with *context* $x^t$, which is text that (1) summarizes the robot observation at each time step (e.g., using a vision-based object detector or an oracle; see Fig. 1) and (2) includes the user instruction. An *augmented context* $\tilde{x}^t$ is obtained by appending the LLM-generated plans to the context $x^t$.

2. Prediction set generation: We use CP to choose a *subset* $C(\tilde{x}^t) \subseteq \mathcal{Y}$ of candidate plans using the LLM's (uncalibrated) confidence $\hat{f}(\tilde{x}^t)_y$ in each prediction $y \in \mathcal{Y}$ given the context $\tilde{x}^t$.

3. Human help: If the prediction set is a non-singleton, the robot leverages help from a human (or any other supervisor agent, denoted as a function $f_{\mathcal{H}}$) to arrive at an unambiguous next step $y_{\mathcal{H}} \in C(\tilde{x}^t)$.

4. Low-level control: A low-level module $\varphi$ converts the plan in $y_{\mathcal{H}}$ to an action $a_t = \varphi(y_{\mathcal{H}})$.

**Goal: uncertainty alignment.** Often in real-world settings, language instructions $\ell$ can be ambiguous, e.g., "place the bowl in the microwave" does not specify that the human may prefer the plastic bowl (Fig. 1). Our goal in this work is to address uncertainty alignment: achieve a desired level of task success while minimizing human help. We formalize this by considering a joint distribution $\mathcal{D}$ over *scenarios* $\xi := (e, \ell, g)$, where $e$ is an environment (POMDP), $\ell$ is a (potentially ambiguous) language instruction, and $g$ is a goal (e.g., formulated as a subset of acceptable states in the POMDP and partially observable through $l$). Importantly, we do not assume knowledge of $\mathcal{D}$, except that we can sample a finite-size dataset of i.i.d. scenarios from it. We formalize uncertainty alignment in our setting as (i) *calibrated confidence:* the robot's policy (with human help as described above) succeeds with a user-specified probability $1-\epsilon$ over new scenarios $\xi \sim \mathcal{D}$, and (ii) *minimal help:* the policy minimizes the number $|C(\cdot)|$ of options presented to the human on average across scenarios $\xi \sim \mathcal{D}$.

## 3  Calibrating LLM Confidence with Conformal Prediction

The MCQA setup above allows us to apply CP to obtain calibrated confidence guarantees while (approximately) minimizing help. We introduce CP below, and then present the different practical settings we consider (possibly involving multiple planning steps and/or multiple correct plans per step).

## 3.1 Background: Conformal Prediction

For now, we drop the timestep superscript and consider a generic MCQA setup with pairs $(\tilde{x}, y)$ consisting of input $\tilde{x}$ and true label $y$. Suppose there is a *calibration* set $Z = \{z_i = (\tilde{x}_i, y_i)\}_{i=1}^N$ of such pairs drawn i.i.d. from an *unknown* distribution $\mathcal{D}$ over $\mathcal{Z} := \mathcal{X} \times \mathcal{Y}$. Now, given a new i.i.d. sample $z_{\text{test}} = (\tilde{x}_{\text{test}}, y_{\text{test}})$ with unknown true label $y_{\text{test}}$, CP generates a *prediction set* $C(\tilde{x}_{\text{test}}) \subseteq \mathcal{Y}$ that contains $y_{\text{test}}$ with high probability [3]:

$$\mathbb{P}\big(y_{\text{test}} \in C(\tilde{x}_{\text{test}})\big) \geq 1 - \epsilon, \tag{1}$$

where $1 - \epsilon$ is a user-defined threshold (desired task success level in our setting) that affects the size of $C(\cdot)$.

To generate $C(\tilde{x}_{\text{test}})$, CP first uses the LLM's confidence $\hat{f}$ (cf. Section 2) to evaluate the set of *nonconformity scores* $\{\kappa_i = 1 - \hat{f}(\tilde{x}_i)_{y_i}\}_{i=1}^N$ over the calibration set — the higher the score is, the less each data in the calibration set *conforms* to the data used for training $\hat{f}$. Then CP performs calibration by defining $\hat{q}$ to be the $\frac{\lceil (N+1)(1-\epsilon) \rceil}{N}$ empirical quantile of $\kappa_1, ..., \kappa_N$. Lastly, CP generates $C(\tilde{x}_{\text{test}}) = \{y \in \mathcal{Y} \mid \hat{f}(\tilde{x}_{\text{test}})_y \geq 1 - \hat{q}\}$, i.e., the prediction set that includes all labels that the predictor is at least $1 - \hat{q}$ confident in. The generated prediction set ensures that the coverage guarantee in Eq. (1) holds.

**Dataset-conditional guarantee.** The probability in Eq. (1) is over *both* the sampling of the calibration set $Z$ and $z_{\text{test}}$ (i.e., a *marginal* guarantee). Thus, to ensure the desired probability of coverage for each new $z_{\text{test}}$, one needs a fresh calibration set. Instead, we apply the following *dataset-conditional* guarantee [9] which is conditioned on a particular calibration dataset being sampled, and thus can be applied to new test data without re-calibration:

$$\mathbb{P}\big(y_{\text{test}} \in C(\tilde{x}_{\text{test}}) \mid \{z_1, ..., z_N\}\big) \geq \text{Beta}_{N+1-v, v}^{-1}(\delta), \quad v := \lfloor (N+1)\hat{\epsilon} \rfloor; \tag{2}$$

it holds with probability $1 - \delta$ over the sampling of $Z$, and $\text{Beta}_{N+1-v, v}^{-1}(\delta)$ denotes the inverse CDF (quantile) level of $\delta$ in a Beta distribution with parameters $N + 1 - v$ and $v$, and $\hat{\epsilon}$ is the threshold used for calibration. In practice, we use a modest-sized calibration dataset ($N = 400$) and $\delta = 0.01$, and adjust $\hat{\epsilon}$ to achieve the desired $1 - \epsilon$ coverage (with probability $1 - \delta = 0.99$ over the sampling of the calibration set).

**Minimal prediction set size.** From [10, Thm. 1], $C(\cdot)$ achieves the smallest average set size among possible prediction schemes $\mathcal{C}$ that achieve the coverage guarantee, if $\hat{f}(\tilde{x})_y$ models true conditional probabilities:

$$\min_{C \in \mathcal{C}} \mathbb{E}_{(\tilde{x}, \cdot) \sim \mathcal{D}} \big[|C(\tilde{x})|\big], \text{ subject to (1)}. \tag{3}$$

The assumption that $\hat{f}$ models true conditional probabilities may be a good approximation for LLMs trained on large-scale data with a proper scoring rule [11]; one can also obtain bounds on near-optimal average set sizes for CP using $\hat{f}$ that approximately model conditional probabilities [12, 10], but we omit these results for brevity. Overall, CP is a powerful and easy-to-use statistical tool to produce (1) tight coverage guarantees— addressing the goal of *calibrated confidence* and (2) small prediction sets for unseen data given a blackbox predictor like a LLM and an unknown data distribution—addressing our second goal of *minimal help*.

## 3.2 Single-Step Uncertainty Alignment

We now demonstrate how to use CP to achieve uncertainty alignment with a user-defined task completion rate $1 - \epsilon$. We first consider a single-step setting, where the LLM plans only once given a context. For simplicity, we again drop the timestep superscript $t$ in this section.

**Data collection.** We collect $N$ i.i.d. scenarios from the distribution $\mathcal{D}$, and the corresponding contexts summarizing the robot observation and instruction. We use the MCQA approach from Section 2 to generate candidate plans and then label each augmented context $\tilde{x}$ (i.e., context combined with plans) with the correct label (here and in Section 3.3, we assume that there is a *unique* correct candidate plan; we provide an extension to multiple acceptable options in Section A2). We thus obtain a calibration set $Z = \{z_i = (\tilde{x}_i, y_i)\}_{i=1}^N$ with pairs of augmented contexts and correct labels.

**Calibration.** Next we follow Section 3.1 to perform calibration: first adjust $\hat{\epsilon}$ to achieve the $1 - \epsilon$ coverage based on Eq. (2) and then find the quantile level $\hat{q}$. Given a new context $\tilde{x}_{\text{test}}$ (after MCQA in a new scenario) at test time, we can construct the calibration set $C(\tilde{x}_{\text{test}})$ that contains $y_{\text{test}}$ with $1 - \epsilon$ probability.

**Triggering help.** If $C(\tilde{x}_{\text{test}})$ is a singleton, the robot executes the corresponding plan. Otherwise, we deem the LLM uncertain over possible actions and trigger human help. The robot presents the human

with $C(\tilde{x}_{\text{test}})$ (including the corresponding plans in text) and asks the human to choose one. The human chooses $y_{\text{test}}$ if $y_{\text{test}} \in C(\tilde{x}_{\text{test}})^2$, or halts the operation otherwise. This setup turns the coverage guarantee from CP to the task completion guarantee:

**Proposition 1 (Single-step uncertainty alignment)** *Consider a single-step setting where we use CP with coverage level $1-\epsilon$ to generate prediction sets and seek help whenever the set is not a singleton. With probability $1-\delta$ over the sampling of the calibration set, the task completion rate over new test scenarios drawn from $\mathcal{D}$ is at least $1-\epsilon$. If $\hat{f}$ models true conditional probabilities, the average prediction set size is minimized among possible prediction schemes that achieve $1-\epsilon$ completion rate.*

The proof immediately follows from the fact that under the assumption of accurate human help, the robot fails only when the prediction set does not contain the true label; the prediction set minimality follows Eq. (3). Thus, our approach addresses the goals of calibrated confidence and minimal help from Section 2.

### 3.3 Multi-Step Uncertainty Alignment

Now we extend the CP-based uncertainty alignment approach to settings where the LLM plans in multiple timesteps. This setting can be helpful when the LLM receives feedback from the environment or human between steps. However, the original CP formulation cannot be applied here since the context $x^t$ between steps are dependent; moreover, the robot's actions at step $t$ influence the distribution over contexts that the robot observes at future steps. Thus, the i.i.d. assumption for the coverage guarantee is no longer valid. Here, we present a novel extension of CP to multi-step settings that tackles this challenge.

**Sequence-level calibration.** The key ideas are to (i) *lift* the data to sequences, and (ii) perform calibration at the sequence level using a carefully designed nonconformity score function (that allows for causal construction of the prediction set at test time). Suppose that each data point consists of a sequence of augmented context $\bar{x} = (\tilde{x}^0, \tilde{x}^1, ..., \tilde{x}^{T-1})$ and true labels $\bar{y} = (y^0, y^1, ..., y^{T-1})$, where $T$ is the time horizon and $\tilde{x}^t$ arises from having performed the *correct* actions in previous steps. The distribution $\mathcal{D}$ over scenarios induces a distribution over data sequences. We can again collect a calibration set $\bar{Z} = \{\bar{z}_i = (\bar{x}_i, \bar{y}_i)\}_{i=1}^N$. Next we use the lowest score over the timesteps as the score for the sequence[3]:

$$\hat{f}(\bar{x})_{\bar{y}} := \min_{t \in [T]} \hat{f}(x^t)_{y^t}. \tag{4}$$

With the standard calibration procedure in Section 3.1, we construct a *sequence-level* prediction set $\overline{C}(\bar{x}_{\text{test}}) := \{\bar{y} \in \mathcal{Y}^T \mid \hat{f}(\bar{x}_{\text{test}})_{\bar{y}} \geq 1 - \hat{q}\}$ for a new context sequence $\bar{x}_{\text{test}}$ with the quantile level $\hat{q}$.

**Causal construction of $C(\bar{x})$ at test time.** Note that $\overline{C}(\bar{x}_{\text{test}})$ is constructed with the full sequence $\bar{x}_{\text{test}}$ at once. However, at test time, we do not see the entire sequence of contexts all at once but rather $x_{\text{test}}^t$ one at a time. We thus need to construct $\overline{C}(\bar{x}_{\text{test}})$ in a *causal* manner (i.e., always relying only on current/past information). Consider the causally constructed prediction set $C^t(x_{\text{test}}^t) := \{y^t \mid \hat{f}(x_{\text{test}}^t)_{y^t} \geq 1 - \hat{q}\}$ at time $t$ using the same quantile level $\hat{q}$ from the non-causal calibration above, and define $C(\bar{x}_{\text{test}}) := C^0(x_{\text{test}}^0) \times C^1(x_{\text{test}}^1) \times \cdots \times C^{T-1}(x_{\text{test}}^{T-1})$. We would like to obtain a lower bound on the sequence-level coverage, $\mathbb{P}(\bar{y}_{\text{test}} \in C(\bar{x}_{\text{test}})) \geq 1-\epsilon$.

**Claim 1** *For any $\bar{y} \in \mathcal{Y}^T$, $\bar{y} \in \overline{C}(\bar{x}_{test}) \iff \bar{y} \in C(\bar{x}_{test})$.*

**Proposition 2 (Multi-step uncertainty alignment)** *Consider a multi-step setting where we use CP with coverage level $1-\epsilon$ to causally construct the prediction set and seek help whenever the set is not a singleton at each timestep. With probability $1-\delta$ over the sampling of the calibration set, the task completion rate over new test scenarios drawn from $\mathcal{D}$ is at least $1-\epsilon$. If $\hat{f}$ models true conditional probabilities, the average prediction set size is minimized among possible prediction schemes that achieve $1-\epsilon$ completion rate.*

The proofs are deferred to Section A1. Claim 1 allows us to construct causal prediction sets from non-causal calibration, and then we show the sequence-level task completion rate guarantee still holds.

**Multiple acceptable options.** Often, there can be multiple acceptable options at the same timestep, e.g., the task is to bring the human a soda, and either the Coke or Sprite on the table is acceptable. In such settings,

---

[2] If the correct option in $C(\tilde{x}_{\text{test}})$ is 'E', the human provides the correct action that was not listed by the robot. The robot continues with this action.

[3] We overload notation here and use $\hat{f}$ to also assign confidence scores to sequences.

we would like the prediction set to contain *at least one* acceptable option. We extend our method and confidence guarantees to this setting for both single- and multi-step problems in Section A2 and Section A3.

# 4 Experiments

We evaluate our framework in a diverse set of language-instructed tasks and environments below, and demonstrate its effectiveness in achieving a user-specified task completion rate while minimizing user help. We use PaLM-2L [13] as the LLM in all examples unless otherwise noted. We parameterize the scenario distribution for each experiment with details shown in Section A6.

**Baselines.** A straightforward way to construct prediction sets given a desired $1-\epsilon$ coverage is to rank options according to confidence and construct a set such that the cumulative confidence exceeds $1-\epsilon$; we consider two baselines that are based on such cumulative thresholding but use different kinds of scores: (1) **Simple Set** uses the same $\hat{f}$ as KNOWNO; (2) **Ensemble Set** [14] instead uses the frequencies of the LLM outputting $y \in \mathcal{Y}$ (out of 20 trials total) with randomized prompting. However, such ways of constructing prediction sets are not guaranteed to achieve $1-\epsilon$ coverage as the probabilities can be miscalibrated [15], and often include additional unnecessary options [16]. Instead of using cumulative thresholding, KNOWNO constructs prediction sets by including options with scores higher than a threshold computed using CP, which results in statistical guarantees. We also introduce two prompt-based baselines: **Prompt Set** prompts the LLM to directly output the prediction set (e.g., "Prediction set: [A, C]"); **Binary** prompts the LLM to directly output a binary indicator for uncertainty (e.g., "Certain/Uncertain: Certain"), which is used in other LLM-based planning work [2] for triggering human intervention. Note that the $\epsilon$ level is not used in Prompt Set or Binary, and so the user cannot explicitly control the task success rate. Lastly, we consider **No Help** where the option with the highest score is always executed without any human intervention.

## 4.1 Simulation: Tabletop Rearrangement

A robot arm is asked to rearrange objects on a table in the PyBullet simulator [17] (Fig. 1 right middle). Each scenario is initialized with three bowls and blocks of green, yellow, and blue colors. The task is to move a certain number of blocks or bowls towards a different object or at a specific location around it. We introduce three settings based on different types of ambiguities in the user instruction: (1) *Attribute* (e.g., referring to the bowl with the word "receptacle"), (2) *Numeric* (e.g., under-specifying the number of blocks to be moved by saying "a few blocks"), and (3) *Spatial* (e.g., "put the yellow block next to the green bowl", but the human has a preference over placing it at the front/back/left/right).

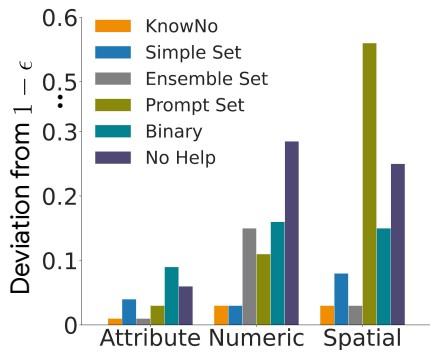

Figure 3: Deviation from specified task success level $1-\epsilon=0.85$ to the empirical success rate for the three settings in **Simulation**. 200 trials are run for each method/setting.

**KNOWNO achieves target task success rate consistently.** First, we investigate whether KNOWNO and the baselines achieve a given target task success rate *consistently* in the three settings — we set the failure level $\epsilon=0.15$. In Fig. 3 we show the difference between achieved and target rates for all methods. Results show that KNOWNO provides task success rates closest to the target overall, due to the coverage guarantee from CP. Prompt Set, Binary, and No Help have larger deviations from the target since the user has no control over the error rate. Also, as the scenarios get increasingly ambiguous (least in Attribute and most in Spatial), the baselines show larger deviations.

**KNOWNO achieves high task success rate with lower human help as $\epsilon$ varies.** In Fig. 4 we vary the target error rate $\epsilon$ and show the full curves of task success rate vs. prediction set size and human help rate averaged over the three settings. For KNOWNO, Simple Set, and Ensemble Set, specifying a lower $\epsilon$ improves the empirical task success rate while also requiring more human intervention. The most natural comparison is between KNOWNO and Simple Set; both use next-token probabilities from the LLM as the confidence score but KNOWNO applies CP and Simple Set applies cumulative thresholding. KNOWNO achieves higher success-to-help ratios across $\epsilon$ levels, thanks to calibrated confidence from CP. Meanwhile, Prompt Set and Binary do not allow controlling success rates. Prompt Set performs the worst, indicating the challenge of prompting-based methods for calibrated prediction set. Binary performs favorably at certain success levels but is (1) inflexible and (2) does not provide human with a prediction set for easier feedback.

In addition, Fig. A10 shows the results for individual ambiguity settings. As the scenarios become more ambiguous, KNOWNO shows a greater reduction of human help compared to Simple Set — as much as 24% at certain success levels.

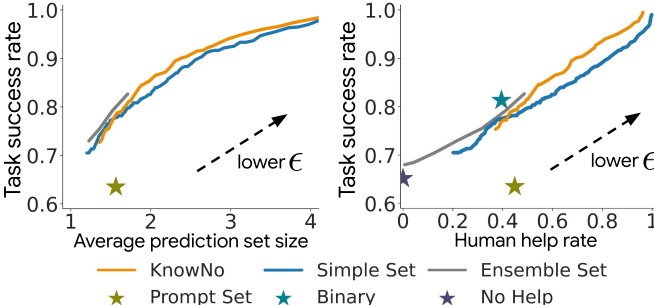

**Ensemble Set can perform well but is computationally expensive.** Fig. 4 also shows that Ensemble Set provides high task success rate with small amount of human help at higher $\epsilon$ levels. However, there are two main drawbacks. First, we find in some scenarios that even with 20 randomized prompts, the LLM can fail to choose the correct option and thus assigns zero probability to it. As shown in Fig. 4, this means that Ensemble Set can fail to improve once it reaches some level of human help. Second, it requires $20\times$ inference time compared to other methods. Investigating how to lower the computational burden and combining ensemble-based probabilities with CP can be a fruitful future direction.

Figure 4: Comparison of task success rate vs average prediction set size (Left) and vs. human help rate (Right) in **Simulation** averaged over the three settings. 200 trials are run for each method. $\epsilon$ is varied from 0.25 to 0.01 for KNOWNO, and from 0.6 to 0.01 for Simple Set and Ensemble Set. Binary and No Help are not shown on the left since prediction sets are not provided.

## 4.2 Hardware: Multi-Step Tabletop Rearrangement

In this example, a real UR5 robot arm is asked to sort a variety of toy food items on a table (Fig. 5). In each scenario, three items are placed on the table initially, and the task is to sort them based on human preferences; we simulate a human with strong preferences for healthy food like eggs and fruits, and dislike for less healthy food like donuts and Skittles candies. To introduce ambiguities, the context for the LLM reveals only a subset of the preferences. Here we consider a multi-step setting with possibly multiple acceptable options per step — the LLM plans the new step conditioned on the previous action taken.

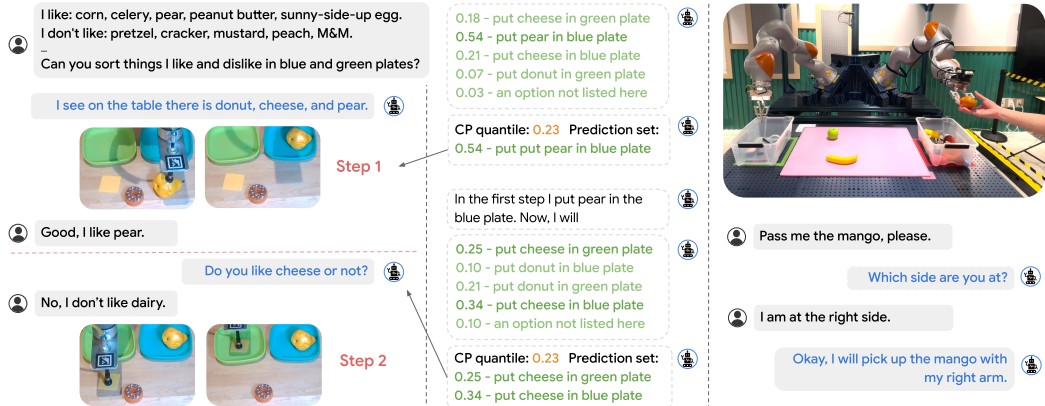

Figure 5: (Left) Multi-step CP is applied in **Hardware Tabletop Rearrangement**. (Right) CP models ambiguity in possible human locations and triggers clarification from the human in **Bimanual**.

**KNOWNO reduces step-wise and trial-wise intervention rate in multi-step setting.** Since Section 4.1 has shown that Ensemble Set can be expensive (even more so in the multi-step setting) and Prompt Set and Binary can fail to achieve the user-specified success level, we focus on comparing KNOWNO with Simple Set for the remainder of the evaluation. Here we set the desired error level $\epsilon = 0.25$. Since Simple Set does not provide coverage, we first find $\epsilon = 0.42$ for

| Method | $1 - \epsilon$ | Plan Succ | Task Succ | Set Size | Help-Step | Help-Trial |
|---|---|---|---|---|---|---|
| KNOWNO | 0.75 | 0.76 | 0.74 | **1.72** | **0.58** | **0.92** |
| Simple Set | 0.58 | 0.76 | 0.72 | 2.04 | 0.72 | 1.00 |
| No Help | - | 0.41 | 0.38 | - | 0 | 0 |

Table 1: Results for **Hardware Multi-Step Tabletop Rearrangement**. Plan success rate is fixed between KNOWNO and Simple Set.

Simple Set that achieves the same planing error rate as KNOWNO in simulation. Then we run 50 trials for both methods with real hardware. Table 1 shows KNOWNO reduces the human help rate by 14% step-wise and 8% trial-wise, while also reducing the average prediction set size. Compared to Simple

Set that uses a much higher $\epsilon$, KNOWNO achieves *trial-level* task success rate precisely leveraging the Multi-Step Uncertainty Alignment from Sec. 3.3. We also find that if we set $\epsilon = 0.25$ for Simple Set, the planner is grossly over-conservative and requires a step-wise help rate of $87\%$.

**Bimanual manipulation.** We additionally present results for a bimanual object rearrangement setup where ambiguities arise from the choice of the arm (Fig. 5 right); results are deferred to Section A4.

### 4.3 Hardware: Mobile Manipulator in a Kitchen

In this example, each scenario involves a mobile manipulator in front of a countertop and next to a set of recycling/compost/landfill bins in a real office kitchen (Fig. 1). The tasks include picking up some object on the counter, and possibly

| Method | Model | $1-\epsilon$ | Plan Succ | Task Succ | Set Size | Help |
|---|---|---|---|---|---|---|
| KNOWNO | PaLM-2L | 0.85 | 0.87 | 0.76 | 2.22 | **0.67** |
| Simple Set | PaLM-2L | 0.76 | 0.87 | 0.75 | 2.38 | 0.81 |
| No Help | PaLM-2L | - | 0.62 | 0.51 | - | 0 |
| KNOWNO | GPT-3.5 | 0.85 | 0.87 | - | 2.50 | 0.86 |

Table 2: Results for **Hardware Mobile Manipulation**. Plan success rate is fixed between KNOWNO and Simple Set.

putting it in the drawer, or disposing of it in one of the bins. For the distribution of possible scenarios, we introduce new types of ambiguities based on Winograd Schemas [18] (e.g., "There is the apple and bottled water on the counter...it is rotten. Can you dispose of it?"), and ones that potentially involve unsafe actions (e.g., "place the bowl in the microwave."; there is a plastic bowl and a metal bowl, but only the plastic one is safe for the microwave). This is a single-step setting with multiple acceptable options. In Table 2, we compare KNOWNO to Simple Set again by first setting $\epsilon = 0.15$ and finding $\epsilon = 0.24$ for Simple Set that achives the same plan success rate. The hardware experiment results again show that KNOWNO reduces the human help rate by $14\%$ and also reduces the average prediction set size.

**Target success guarantee from KnowNo is robust to varying LLM choice.** We also run KNOWNO with GPT-3.5 (*text-davinci-003*) from OpenAI (without hardware evaluation). However, we find that it exhibits significant MCQA bias towards options D and E and against A and B, affecting the overall performance. Nonetheless, KnowNo still achieves $1-\epsilon$ target success rate, as the coverage guarantee from CP makes no assumption about the LLM confidences (e.g., calibrated or accurate) — KnowNo flexibly compensates for the degraded LLM performance by triggering more human intervention.

## 5 Related Work

**Uncertainty quantification for LLMs.** Motivated by LLMs' overconfidence and hallucination, there has been a growing body of work in quantifying and better calibrating uncertainty [19, 20, 21, 22, 23, 24, 25]. In contrast to typical calibration methods that associate uncertainty with point-valued outputs, CP-based methods for language modeling provide coverage guarantees for set-valued predictors [26, 27, 28, 29].

**Conformal prediction in robotics.** To the best of our knowledge, this work is the first to employ CP for language-based planning. Prior work has utilized CP for fault detection, trajectory prediction, and planning in dynamic environments [30, 31, 32, 33]. We provide a novel multi-step extension to CP to guarantee correctness for the entire planning horizon by performing sequence-level calibration in settings where the robot's actions influence the distribution of future inputs.

**Human-robot dialogue and interaction.** KNOWNO builds off previous work in robotics that addresses effective human-robot interaction through dialogue [34, 35, 36, 37, 38]. KNOWNO uses a relatively simple setup such that the human specifies potentially ambiguous instruction and clarifies it when necessary; such setup does not consider human providing information related to robot observation or possible human error.

## 6 Discussion

**Summary:** We propose KNOWNO, a framework that applies conformal prediction (CP) to address the problem of uncertainty alignment for language-instructed robots, which we formalize as providing statistical guarantees of task completion while minimizing human help. Experiments across a variety of simulated and hardware setups demonstrate that KNOWNO achieves user-specified task completion levels consistently while reducing human help by $10-24\%$ compared to baseline approaches that lack formal assurances.

**Limitations and future work:** The primary limitation of our work is that the task completion guarantee assumes environments (objects) are fully grounded in the text input to the LLM, and the actions proposed by the LLM planner can be executed successfully. In the future, we are looking to incorporate uncertainty of the perception module (e.g., vision-language model) and the low-level action policy (e.g., language-conditioned affordance prediction) into the CP calibration.

**Acknowledgments**

This work was partially supported by the NSF CAREER Award [#2044149] and the Office of Naval Research [N00014-23-1-2148]. We thank Chad Boodoo for helping set up the UR5 hardware experiments, and Jensen Gao, Nathaniel Simon, and David Snyder for their helpful feedback on the paper.

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

## A1    CP in Multi-Step Setting

**Algorithm 1** Multi-step LLM planning with human help.

1: **for** time $t \leftarrow 0$ to $T-1$ **do**
2:      Observe input $x_{\text{test}}^t$
3:      Predict a set $C(x_{\text{test}}^t)$
4:      **if** $C(x_{\text{test}}^t)$ is a singleton **then**
5:          execute corresponding action
6:      **else**
7:          ask for help, which is always assumed to provide the correct label, or the human provide clarification only
8:      **end if**
9: **end for**

**Proof of Claim 1:** Suppose $\overline{y} \in \overline{C}(\overline{x}_{\text{test}})$. We have,

$$\overline{y} \in \overline{C}(\overline{x}_{\text{test}}) \Longleftrightarrow \min_t \hat{f}(x_{\text{test}}^t)_{y^t} \geq 1-\hat{q} \tag{A1}$$

$$\Longleftrightarrow \hat{f}(x_{\text{test}}^t)_{y^t} \geq 1-\hat{q}, \qquad \forall t \in [T] \tag{A2}$$

$$\Longleftrightarrow y^t \in C^t(x_{\text{test}}^t), \qquad \forall t \in [T] \tag{A3}$$

$$\Longleftrightarrow \overline{y} \in C(\overline{x}_{\text{test}}). \tag{A4}$$

**Proof of Proposition 2:** Since we can bound the probability that $\overline{y}_{\text{test}} \notin \overline{C}(\overline{x}_{\text{test}})$, we can also bound the probability that $\overline{y}_{\text{test}} \notin C(\overline{x}_{\text{test}})$. From the conformalization procedure, we have the following *dataset-conditional* guarantee: with probability $1-\delta$ over the sampling of the calibration set $\overline{Z}$, we have

$$\mathbb{P}(\overline{y}_{\text{test}} \in \overline{C}(\overline{x}_{\text{test}})|\overline{Z}) \geq \text{Beta}_{N+1-v,v}^{-1}(\delta), \quad v = \lfloor (N+1)\hat{\epsilon} \rfloor \tag{A5}$$

$$\overset{\text{Claim 1}}{\Longrightarrow} \mathbb{P}(\overline{y}_{\text{test}} \in C(\overline{x}_{\text{test}})|\overline{Z}) \geq \text{Beta}_{N+1-v,v}^{-1}(\delta), \tag{A6}$$

where $\hat{\epsilon}$ is chosen such that $\epsilon = 1 - \text{Beta}_{N+1-v,v}^{-1}(\delta)$. Hence, the following *marginal* guarantee also holds:

$$\mathbb{P}(\overline{y}_{\text{test}} \in \overline{C}(\overline{x}_{\text{test}})) \geq 1-\hat{\epsilon}$$

$$\overset{\text{Claim 1}}{\Longrightarrow} \mathbb{P}(\overline{y}_{\text{test}} \in C(\overline{x}_{\text{test}})) \geq 1-\hat{\epsilon}.$$

This result provides a bound on the task completion rate if $\overline{x}_{\text{test}}$ is drawn using the distribution $\mathcal{D}$. However, recall that the sequence $\overline{x}$ of augmented contexts as defined in Section 3.3 arises from having performed the *correct* actions in previous steps; incorrect actions may result in a distribution shift. In order to obtain a bound on the task completion rate, we consider three cases at any given timestep: (1) the prediction set is a singleton and contains the correct label, (2) the prediction set is not a singleton but does contain the correct label, and (3) the prediction set does not contain the true label. The robot performs the correct action in the first two cases (without help in (1) and with help in (2)), while CP bounds the probability of case (3). Thus, the CP bound translates to a bound on the task success rate.

As seen in Eq. (3), we have from [10, Thm. 1], that we achieve the smallest average set size among all possible sequence-level prediction schemes, $\overline{\mathcal{C}}$, if $\hat{f}$ models the prediction uncertainty accurately,

$$\min_{\overline{C} \in \overline{\mathcal{C}}} \mathop{\mathbb{E}}_{(\overline{x}, \cdot) \sim \mathcal{D}} \left[ |\overline{C}(\overline{x})| \right], \text{ subject to } \mathbb{P}(\overline{y} \in \overline{C}(\overline{x})) \geq 1-\hat{\epsilon}. \tag{A7}$$

## A2    CP in Settings with Multiple Acceptable Options Per Step

**Proposition 3 (Multi-label uncertainty alignment)** *Consider a setting where we use CP with coverage level $1-\epsilon$ to construct the prediction set when there are multiple true labels and seek help whenever the set is not a singleton at each timestep. With probability $1-\delta$ over the sampling of the calibration set, the task completion rate over new test scenarios drawn from $\mathcal{D}$ is at least $1-\epsilon$.*

**Proof:** We have a dataset of $Z = \{(\tilde{x}_i, Y_i), ...\}_{i=1}^N$ sampled i.i.d. from a data distribution $\mathcal{D}$ for calibration (we use the same notation $\mathcal{D}$ as in the single-label setting here), where $Y_i := \{y_{i,j}\}_{j=1}^{J_i}$ is the set of true labels for a single trial. For each label, we use the same heuristic notion of confidence, $\hat{f}(x)_y \in [0,1]$.

We define an operator $\beta: \mathcal{X} \times \mathcal{Y}^J \to \mathcal{Y}$ where $\mathcal{X}$ is the space of contexts and $\mathcal{Y}$ is the space of labels:

$$\beta(x,Y) := \underset{y \in Y}{\operatorname{argmax}} \hat{f}(x)_y, \tag{A8}$$

which takes the true label with the highest confidence value from the true label set.

If we consider applying $\beta$ to every point in the support of $\mathcal{D}$, a new distribution $\mathcal{D}'$ is induced. We also consider the induced dataset of samples $S' = \{(x_i, y_i^{\max})\}_{i=1}^N$, where $y_i^{\max} := \beta(x_i, Y_i)$. Then we can perform the usual conformalization and obtain the guarantee that with

$$C(x_{\text{test}}) := \{y | \hat{f}(x_{\text{test}})_y \geq 1 - \hat{q}\}, \tag{A9}$$

the following *marginal* guarantee holds,

$$\mathbb{P}(y_{\text{test}}^{\max} \notin C(x_{\text{test}})) \leq \hat{\epsilon}, \tag{A10}$$

$$\Rightarrow \mathbb{P}(\underset{y \in Y_{\text{test}}}{\operatorname{argmax}} \hat{f}(x_{\text{test}})_y \notin C(x_{\text{test}})) \leq \hat{\epsilon}, \tag{A11}$$

$$\Rightarrow \mathbb{P}(\beta(x_{\text{test}}, Y_{\text{test}}) \notin C(x_{\text{test}})) \leq \hat{\epsilon}, \tag{A12}$$

and the following *dataset-conditional* guarantee holds when we choose $\hat{\epsilon}$ such that $\epsilon = 1 - \text{Beta}_{N+1-v,v}^{-1}(\delta)$ where $v = \lfloor (N+1)\hat{\epsilon} \rfloor$,

$$\mathbb{P}(\beta(x_{\text{test}}, Y_{\text{test}}) \in C(x_{\text{test}}) | Z) \geq 1 - \epsilon. \tag{A13}$$

Hence, $C(x_{\text{test}})$ contains the true label with the highest confidence with probability at least $1 - \epsilon$.

At test time, we sample $(x_{\text{test}}, Y_{\text{test}})$ from $\mathcal{D}$ that is i.i.d. with samples in $S$ — for the guarantee to hold for $\beta(x_{\text{test}}, Y_{\text{test}})$, we need to show $\beta(x_{\text{test}}, Y_{\text{test}})$ is a sample from $\mathcal{D}'$ that is i.i.d. with samples in $S'$. This is true since functions of independent random variables are independent, and functions of identically distributed random variables are identically distributed if the functions are measurable.

### A3 CP in Multi-Step Setting with Multiple Acceptable Options Per Step

**Proposition 4 (Multi-step, multi-label uncertainty alignment)** *Consider a multi-step setting where we use CP with coverage level $1 - \epsilon$ to causally construct the prediction set when there may be multiple true labels at any step and seek help whenever the set is not a singleton at each timestep. With probability $1 - \delta$ over the sampling of the calibration set, the task completion rate over new test scenarios drawn from $\mathcal{D}$ is at least $1 - \epsilon$.*

**Proof:** For the multi-step setting, each trial now involves a sequence of contexts $\bar{x}$ and a set of sequences of true labels:

$$\overline{Y} = \{\bar{y}_1, \bar{y}_2, ..., \bar{y}_M\}, \tag{A14}$$

where $\bar{y}_m := (y_m^0, y_m^2, ..., y_m^{T-1})$. For example, $\overline{Y}$ can contain the sequence of "blue block, yellow block, green block", "green block, blue block, yellow block", ..., for the task of picking up three blocks. We collect a dataset of $\overline{Z} = \{(\bar{x}_i, \overline{Y}_i)\}$ of i.i.d. samples from the data distribution $\overline{\mathcal{D}}$.

Unlike the single-step setting, here we cannot apply $\beta$ to the set of true labels in each step since we are reasoning over a *set of sequences*, and not a *sequence of sets* of true labels. Notably, the true label set at time step $t$ depends upon the sequence of previously chosen true labels.

Let $Y^t[x^0, \bar{y}^{t-1}]$ denote the set of true labels at timestep $t$, *conditioned* upon the initial context $x^0$ and a partial sequence of past true labels $\bar{y}^{t-1} := (y^0, ..., y^{t-1})$ extracted from $\overline{Y}$. We then autoregressively define the following sequence:

$$\overline{\beta}_0(\bar{x}, \overline{Y}) := \underset{y \in Y^0}{\operatorname{argmax}} \hat{f}(x^0)_y, \quad Y^0 := \{y_1^0, ..., y_M^0\} \tag{A15}$$

$$\overline{\beta}_t(\bar{x}, \overline{Y}) := \overline{\beta}_{t-1}(\bar{x}, \overline{Y}) \bigcup \underset{y \in Y^t[x^0, \overline{\beta}_{t-1}(\bar{x}, \overline{Y})]}{\operatorname{argmax}} \hat{f}(x^t)_y, \quad t = 1, ..., T-1. \tag{A16}$$

For convenience, we denote $\overline{\beta}_t(\bar{x}, \overline{Y})[\tau]$ the $\tau$ element in $\overline{\beta}_t(\bar{x}, \overline{Y}), \tau \leq t$. An intuitive interpretation is that, we can consider $\overline{Y}$ forming a tree of valid executions (all possible actions that can be taken by choosing each of true labels). Hence, at each time step $t$, $\overline{\beta}_t(\bar{x}, \overline{Y})$ prunes the tree to a single branch by taking the true label with the highest heuristic value $\hat{f}(x^t)$. This reduces the tree of all possible sequences of true

labels to a single branch of true labels with highest confidence. Given this single branch of true labels, we can now perform CP as shown in the multi-step setting in Section A1.

We apply $\overline{\beta}_{T-1}$ to every point in the support of $\overline{\mathcal{D}}$, and a new distribution $\overline{\mathcal{D}}'$ is induced. We consider $\overline{S}' = \{(\overline{x}_i, \overline{y}_i^{\max})\}$, where $\overline{y}_i^{\max} := \overline{\beta}_{T-1}(\overline{x}_i, \overline{Y}_i)$. Let $\overline{Y}_{\text{test}}$ be the set of sequences of true labels for $\overline{x}_{\text{test}}$. Suppose we get the *marginal* bound with $\overline{\beta}_{T-1}$ as the labels:

$$\mathbb{P}(\overline{\beta}_{T-1}(\overline{x}_{\text{test}}, \overline{Y}_{\text{test}}) \notin C(\overline{x}_{\text{test}})) \le \hat{\epsilon}, \tag{A17}$$

and *dataset-conditional* bound when we choose $\hat{\epsilon}$ such that $\epsilon = 1 - \text{Beta}_{N+1-v,v}^{-1}(\delta)$ where $v = \lfloor (N+1)\hat{\epsilon} \rfloor$,

$$\mathbb{P}(\overline{\beta}_{T-1}(\overline{x}_{\text{test}}, \overline{Y}_{\text{test}}) \notin C(\overline{x}_{\text{test}}) | \overline{Z}) \le \epsilon, \tag{A18}$$

which states that at test time, given a context sequence $\overline{x}_{\text{test}}$, we produce a prediction set of sequences; if we consider a sequence consisting of the true label with the highest score at each step, the probability of this sequence covered by $C(\overline{x}_{\text{test}})$ is lower bounded by $1 - \epsilon$. However, we need to be careful of following $\overline{\beta}_t$ at each step at test time. Conside the three cases:

- (1) At a given time-step, the prediction set $C^t(x_{\text{test}}^t)$ does not contain the true label, $\overline{\beta}_t(\overline{x}, \overline{Y})[t]$.

- (2a) The prediction set is a singleton and does contain the true label.

- (2b) The prediction set is not a singleton (but does contain the correct label).

We already bound the probability of (1) happening with the CP bound; (2a) is fine since the LLM will take the correct action; (2b) is more challenging — in this case the robot asks the human for help, and we need to make sure the human "follows" the true label, by choosing the true label in the prediction set with the highest confidence by $\hat{f}$. In practice, we present the labels ranked by $\hat{f}$ and ask the human to choose the true label with the highest rank.

Now let's derive the bound in Eq. (A17) and Eq. (A18). Again we need to consider the causal construction issue. As seen in Section 3.3, we construct the prediction set $\overline{C}(\overline{x}_{\text{test}})$ non-causally using the score function $s_i = 1 - \hat{f}(\overline{x}_i)_{\overline{y}_i^{\max}}$ (taking minimum over steps). For a test sequence $\overline{x}_{\text{test}}$, we apply $\overline{\beta}_{T-1}$ to the true label set of sequences $\overline{Y}_{\text{test}}$ to get $\overline{y}_{\text{test}}^{\max} = \overline{\beta}_{T-1}(\overline{x}_{\text{test}}, \overline{Y}_{\text{test}})$. Now suppose $\overline{y}_{\text{test}}^{\max} \in \overline{C}(\overline{x}_{\text{test}})$, then we can show $\overline{y}_{\text{test}}^{\max} \in C(\overline{x}_{\text{test}})$ with the same proof as the single-label setting, which gives us the bound.

Lastly we need to show the sampled test sequence from $\overline{D}$ leads to a sample from $\overline{\mathcal{D}}'$ i.i.d. with $\overline{S}'$. This is true with the same argument that functions of independent random variables are independent.

## A4    Additional Experiment Setting: Hardware Bimanual Setup

In this example, a real bimanual setup with two Kuka IIWA 7 arms move objects on the table, with one bin at each side (Fig. 5 right). The reachable workspace of each arm is limited so that one arm cannot reach the other end of the table or the other bin. Thus, there can be ambiguities in the choice of the arm depending on the task; e.g., Fig. 5 shows the human asking the robot to pass over the mango, but not specifying where the human is. KNOWNO is able to capture such ambiguities and triggers clarification. We design a scenario distribution (single-step with single acceptable options) with all instructions being ambiguous (thus requiring high human intervention rate): with $\epsilon = 0.15$, the robot achieves $84\%$ plan success with $92\%$ help. With 10 real trials, the robot succeeds 9 times while triggering help for 9 times. Details of the scenario distribution are shown in Section A6.

## A5    LLM Prompt Setup

Next we detail the LLM prompt setup for MCQA applied in KNOWNO. We will use the Mobile Manipulation setting from Section 4.3 as the example.

**Multiple choice generation.**    Given a scenario, we first prompt the LLM to generate four options for possible actions. We apply few-shot prompting as shown in Fig. A1 below with zero temperature. In this scenario, there is a Coke, a bottled tea, and a Pepsi on the counter, and the task is to put the Coke in the top drawer but the choice of drawer is under-specified ("Put the Coke in the drawer please.").

```
We: You are a robot operating in an office kitchen. You are in front of a counter with two closed drawers,
a top one and a middle one. There is also a landfill bin, a recycling bin, and a compost bin.

We: On the counter, there is an orange soda, a Pepsi, and an apple.
We: Put that drink in the top drawer.
You:
A) open the top drawer and put the orange soda in it
B) open the middle drawer and put the Pepsi in it
C) open the middle drawer and put the orange soda in it
D) open the top drawer and put the Pepsi in it

We: On the counter, there is an energy bar, a banana, and a microwave.
We: Put the snack next to the microwave.
You:
A) pick up the energy bar and put it next to the microwave
B) pick up the banana and put it next to the energy bar
C) pick up the banana and put it next to the microwave
D) pick up the energy bar and put it next to the banana

We: On the counter, there is a Coke, a Sprite, and a sponge.
We: Can you dispose of the can? It should have expired.
You:
A) pick up the sponge and put it in the landfill bin
B) pick up the Coke and put it in the recycling bin
C) pick up the Sprite and put it in the recycling bin
D) pick up the Coke and put it in the landfill bin

We: On the counter, there is a bottled water, a bag of jalapeno chips, and a bag of rice chips.
We: I would like a bag of chips.
You:
A) pick up the bottled water
B) pick up the jalapeno chips
C) pick up the kettle chips
D) pick up the rice chips
(The correct option is either B or D, since either jalapeno chips or rice ships are fine.)

We: On the counter, there is a Coke, a bottled unsweetened tea, and a Pepsi.
We: Put the Coke in the drawer please.
You:
```

Figure A1: Prompt used for multiple choice generation in the Mobile Manipulation setting.

After the LLM generates four options, we append an additional option 'an option not listed here' to the four generated ones and then randomize the order to further prevent bias. We then use a zero-shot prompt in Fig. A2 for querying next-token probabilities ('A', 'B', 'C', D, 'E').

```
We: You are a robot operating in an office kitchen. You are in front of a counter with two closed drawers,
a top one and a middle one. There is also a landfill bin, a recycling bin, and a compost bin.

We: On the counter, there is a Coke, a bottled unsweetened tea, and a Pepsi.
We: Put the Coke in the drawer please.
You:
A) pick up the coke
B) pick up the coke and put it in the top drawer
C) pick up the coke and put it in the bottom drawer
D) a different option not listed here
E) pick up the pepsi
We: Which option is correct? Answer with a single letter.
You:
```

Figure A2: Prompt used for next-token prediction with generated multiple choices in the Mobile Manipulation setting.

## A6    Additional Experiment Details

**Environments.**    In addition to Fig. 1 and Fig. 5, here Fig. A3 shows the office kitchen environment with the set of drawers and bins used in the Mobile Manipulator experiments (left), and the bimanual setup with the set of objects used on the mat (right). There is another set of drawers used in the mobile manipulation experiments underneath a much bigger countertop not shown here.

**Scenario Distribution and Calibration Dataset.** Next, we provide more details on the parameterization of the scenario distribution in each experiment setting, in particular, the possible ambiguities with the instruction and goal. With the distributions set up, the calibration dataset are then generated by randomly sampling 400 i.i.d. scenarios from them:

- Simulated setting:
    - Environment: there are always three blocks and bowls of color red, yellow, and green with random locations on the table.
    - Goal: we use the following template: {put, place, move} {a, one, a single of, two, a pair of, three, all, red, yellow, green} {block(s),  bowl(s)} {on,  to the left of,  to the right of,  to the front of,  at

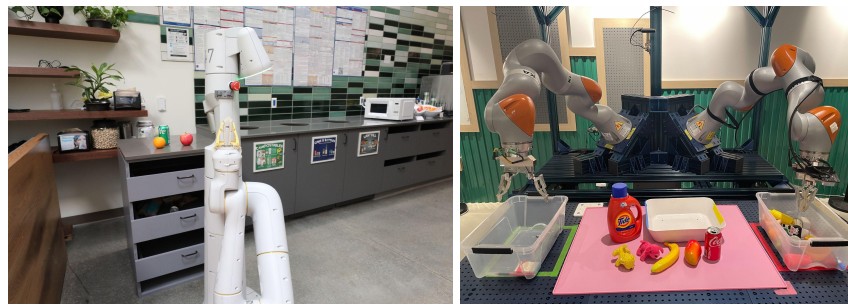

Figure A3: (Left) Office kitchen environment with drawers and bins for the Mobile Manipulation setting. (Right) Bimanual setup with the set of objects used in the experiments.

the back of} the {red, green, yellow} {block(s), bowl(s)}. The scenario distribution is uniform over these possibilities.

- Instruction: for the language instructions, we modify the goal (from the template) according to the ambiguity type. The scenario distribution is uniform over the listed ambiguous cases in each ambiguity type.
    * Attribute ambiguities: refer to the block as one of "cube", "cuboid", "box", "square object", to the bowl as one of "container", "round object", "receptacle", or to either block or bowl as one of "object", "item", "thing" ("move the blue object in yellow bowl"); refer to "blue" as one of "cyan", "navy", to "green" as one of "greenish", "grass-colored", and to "yellow" as "orange" or "gold". This setting is the least ambiguous one among the three ambiguity types.
    * Numeric ambiguities: refer to either two or three numerically with one of "a few", "a couple of", "some", "a handful of" ("put some blocks in the green bowl").
    * Spatial ambiguities: refer to any of the four possible directions with "near", "close to", "beside", "next to", refer to either left to right with "lateral to", and refer to either front or behind with "along the line of sight". This setting is the most ambiguous one among the three ambiguity types.

- Hardware Tabletop Rearrangement setting:
    - Environment: there are three items to be sorted placed randomly on the table, and there is a blue plate and a green plate. 28 toy items are split (Fig. A7) into two categories of human liking them or disliking them: the things the human likes include corn, avocado, celery, carrot, tomato, lettuce, apple, orange, pear, lemon, peanut butter, sunny-side-up egg, egg, and pea; the human dislikes pretzel, cracker, waffle, mustard, ketchup, pizza, meat patty, cheese, chicken drumstick, peach, mango M&M, Skittles, and donut. The three items on the table are uniformly sampled ones from the two categories (1 from likes and 2 from dislikes, or 2 from likes and 1 from dislikes).
    - Goal: place the items that human likes in the blue plate, and ones that human dislikes in the green plate.
    - Instruction: to make the goal ambiguous, only a subset of human preferences are revealed in the prompt:"I like: corn, celery, pear, peanut butter, sunny-side-up egg. I don't like: pretzel, cracker, mustard, peach, M&M". And then the instruction includes "Can you sort things I like and dislike in blue and green plates?"

- Hardware Mobile Manipulator setting:
    - Environment: the full list of possible objects include: bottled water, bottled tea, orange soda, RedBull, Coke, Pepsi, Sprite, rice chips, jalapeno chips, kettle chips, multigrain chips, apple, orange, energy bar, clean sponge, dirty sponge, metal bowl, plastic bowl. Depending on the ambiguity listed below, there is three objects placed on the top of the counter (including randomly sampled distractors from the list). There is also a set of landfill, compost, and recycling bins, a microwave, and a portable stove.
    - Instruction: for convenience, we introduce the possible instructions first in different ambiguous scenarios; they each correspond to possible goals. Please refer to `https://robot-help.github.io/prompts/mobile_tasks.txt` for the full list. The possible instructions are a uniform distribution over different types: (1) single-label, e.g., 'Bring me a Coke' (unambiguous); (2) creative-single-label, e.g., 'I want a healthy fruit to munch on.' which means the apple (unambiguous); (3) multi-label, e.g., 'Bring me a cola.' and either Coke or Pepsi is acceptable; (4)

creative-multi-label, e.g., 'Bring me something with a kick.' and either RedBull or jalapeno chips are acceptable; (5) spatially-ambiguous, e.g., 'Put the Coke in the drawer' or 'Put the Coke near the fruit' which under-specifies the drawer or fruit; (6) unsafe, e.g., 'Can you dispose of the bottle drink? It should have expired.' or 'Place the bowl on the stove, please.'; (7) Winograd, e.g., 'There is a sponge and a bag of rice chips...I don't want to use it for cleaning any more. Can you please dispose of it?" We use the GPT-4 model for generating the creative tasks.

- Goal: the corresponding goal for the ambiguous instructions above. For example, the instruction is "Put the Coke in the drawer", and the goal is uniform over the two possibilities: put the Coke in the top drawer, and put the Coke in the bottom drawer.

- Bimanual setting:

  - Environment: the full list of possible objects include: Coke can, Sprite can, green apple, banana, mango, red block, green block, yellow block, pink plushie, yellow plushie, purple plushie. Depending on the ambiguity listed below, there is three objects placed on the table (including randomly sampled distractors from the list). There is possibly a human standing at the left or right side of the table.

  - Instruction: Please refer to https://robot-help.github.io/prompts/bimanual_tasks.txt for the full list of scenarios sampled from the distribution. The possible instructions are a uniform distribution over different types: (1) 'Pick up the {object} and pass it to me. I am next to the bin.' (2) 'Pick up the {object} with the left arm.' (3) 'Put the {object} in the bin closer to it.' (4) 'Pick up the {object} with the arm closer to it.' (5) 'Pick up the {object}.' (6) 'Pick up the {object} at the handle.' (7) 'Move the {object} to the front of the table.' (8) 'Move the {object} on the sticky rubber mat to the front of the table.'

  - Goal: the corresponding goal for the ambiguous instructions above. For example, the instruction is "Pick up the Coke can and pass it to me. I am next to the bin.", and the goal is pick up the Coke can with the left arm and pass it to human, if human is at the left side, and with the right arm if human is at the right side.

```
# Define possible environment, goal, and instruction
possible_envs = ['three blocks and three bowls']
possible_goals_with_instructions =
{'put three blocks in the green bowl': ['put three blocks in the green bowl', 'put all blocks in the green
bowl', 'put three cubes in the green bowl', …],
 'put two blocks in the blue bowl': ['put two blocks in the blue bowl', 'put a pair of blocks in the blue
bowl', …],
…} # key: goal; value: list of possible instructions

# Sample scenarios by uniformly sampling from possible environments, goals, and instructions
num_calibration_data = 100
calibration_dataset = []
for _ in range(num_calibration_data):
  env = random.choice(possible_envs)
  goal = random.choice(list(possible_goals_with_instructions.keys()))
  instruction = random.choice(possible_goals_with_instructions[goal])
  calibration_dataset.append((env, goal, instruction))
```

Figure A4: Sample algorithm for defining the scenario distribution and generating the calibration dataset in the Simulation setting.

Next we provide more details on some of the baselines.

**Baselines - Ensemble Set.** Our ensemble-based method is a weaker method than the traditional model-based ensemble where multiple copies of neural network are trained and inferred with; however, this is infeasible with the LLM we use. In our work, we randomize over the few-shot examples in the prompt as the ensemble. We select a pool of 20 possible MCQA examples (see examples in Fig. A1), and then randomly sample a certain amount from it for each inference. Note that in this case, Ensemble Set actually has advantage over KNOWNO and Simple Set that, for the same data, it has seen many more examples than the fixed ones in the prompt used in KNOWNO and Simple Set. We only apply ensemble for next-token prediction; the same set of multiple choices generated is used.

**Baselines - Prompt Set.** First, multiple choices are generated in the same way as KNOWNO. Then LLM is prompted to generate the prediction set, with few-shot examples in the prompt showing the possible labels (Fig. A5). For example, "We: Which options are possibly correct? You: A, C, D.".

**Baselines - Simple Set.** Instead of generating multiple choices, the LLM is first prompted to give the most likely action (e.g., "We: Put the Coke can in the drawer. You: I will" shown in Fig. A8). Then we attach the generated response to the same prompt, and ask LLM to label "Certain/Uncertain:" given few-shot examples (Fig. A6).

```
...
We: On the table there are these objects: yellow bowl, blue bowl, blue block, green block, green bowl,
yellow block.
We: Now, Put the blue object at the right side of the grass-colored square object.
You: These are some options:
A) an option not listed here
B) put blue block to the right of green block
C) put blue block to the left of green bowl
D) put blue bowl to the right of green block
E) put blue block to the left of green block
Me: Which options are possibly correct?
You: B, D

We: On the table there are these objects: blue bowl, green block, yellow bowl, yellow block, blue block
green bowl.
We: Now, Put the green round object at the left side of the greenish block.
You: These are some options:
A) put green bowl to the left of yellow block
B) an option not listed here
C) put green bowl to the left of green block
D) put green bowl to the left of blue bowl
E) put green bowl to the left of blue block
Me: Which options are possibly correct?
You: C

We: On the table there are these objects: yellow block, yellow bowl, blue block, green block, blue bowl
green bowl.
We: Now, put the cyan bowl behind the gold object.
You: These are some options:
A) put blue bowl behind yellow block
B) put blue bowl behind green block
C) put blue bowl behind yellow bowl
D) an option not listed here
E) put blue bowl behind blue block
We: Which options are possibly correct?
You:
```

Figure A5: Prompt with few-shot examples (two shown) for LLM generating the prediction set in Prompt Set baseline.

```
...
We: On the table there are these objects: blue bowl, green block, yellow bowl, yellow block, blue block,
green bowl.
We: Now, Put the green round object at the left side of the greenish block.
You: I will put green bowl to the left of yellow block.
Certain/Uncertain: Uncertain

We: On the table there are these objects: blue block, yellow bowl, green block, green bowl, blue bowl,
yellow block.
We: Now, Put the yellow box in front of the cyan receptacle.
You: I will put yellow block at the front of blue block
Certain/Uncertain: Certain

We: On the table there are these objects: blue bowl, blue block, green bowl, yellow bowl, green block,
yellow block.
We: Put the yellow box at the right side of the navy receptacle.
You: I will put yellow block to the right of blue block.
Certain/Uncertain:
```

Figure A6: Prompt with few-shot examples (two shown) for LLM expressing binary uncertainty in Binary baseline.

## A7    Additional Implementation Details

While the focus of KNOWNO is mainly on providing uncertainty alignment for the LLM-based planner, below we provide details of the perception and action modules applied in all examples.

**Perception.**    For all settings except for the Mobile Manipulator, we use either MDETR [39] (UR5 tabletop setting) or Owl-ViT [40] (Simulated and Bimanual settings) open-vocabulary object detector for recognizing the objects in the environment and obtaining the object locations for low-level action. In Simulated and Bimanual settings, the variations of the object types are limited, and with general prompting, the objects are detected without issue. In the UR5 tabletop setting, since we are use a wide variety of toy items (Fig. A7 right), the detector has issues often differentiating objects like peanut butter and meat patty that are both darker colors. We modify the scenario distributions to avoid using such items together in one scenario. In addition, we apply the Segment Anything model [41] to extract the object segmentation masks (shown overlaid in Fig. A7 left), and then use the `polylabel` algorithm [42] to find the most distant internal point of the mask as the suction point (shown as red dots).

**Low-level action.**    In Simulated setting and UR5 tabletop setting, simple pick-and-place actions are executed based on object locations and solving the inverse kinematics. In Bimanual setting, the reachability of the Kuka arm is limited, and the pick-and-place action trajectories are solved using Sequential Quadtratic Programming (SQP) instead [43]. In the Mobile Manipulator setting, for most of the tasks that involve simple pick-and-place and opening the drawers, the action is from an end-to-end policy from the RT-1

```

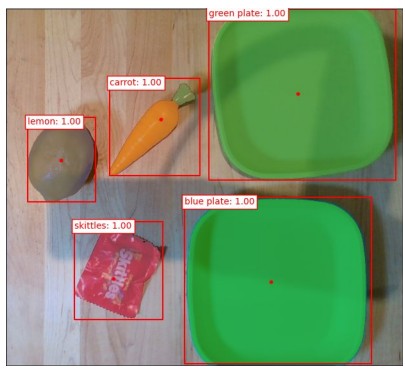
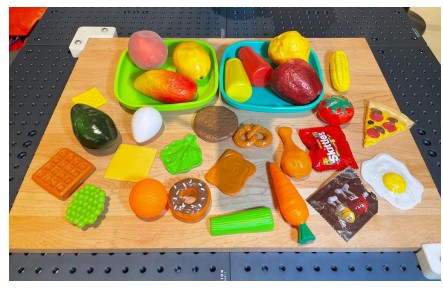

Figure A7: (Left) MDTER [39] object detection with Segment Anything [41] and most distant internal point (red dots) for UR5 tabletop setting. (Right) The total 28 toy items used for the experiments.

policy (please refer to [44] for details), which takes in the raw observation. For some of the hard tasks such as putting the plastic bowl in the microwave and putting the metal bowl on the bowl, object locations are assumed known and we use scripted action policies.

**Human feedback.** In KNOWNO, once human help is triggered, human is presented with the prediction set to choose the correct action from (if there is one). For example, the prediction set could include 'A) put peanut butter in blue plate' and 'C) put peanut butter in green plate' in Hardware Tabletop Rearrangement. In practice, we can convert the prediction set to a question in more natural language, e.g., "Do you like peanut butter or not?" using simple heuristics. In Mobile Manipulation and Bimanual, we prompt the LLM to generate the question based on the prediction set.

## A8 Additional Discussions

**Sentence-level score leads to worse performance.** In Section 2 we hypothesize that the distribution of probabilities of LLM outputs $p(y)$ is highly sensitive to the output length. Here we explore the effect of using sentence output and the perplexity score for CP in Simulation. We still apply multiple choice generation first to obtain the possible options from LLM, and then query LLM scoring, for example, the probability of "put the blue block in the green bowl" with the prompt ending with "I will" (Fig. A8). We test two different method: CP with RawProb, which uses the raw product of conditional probabilities $p(y) = \prod_{i=1}^{k} p(\sigma_i | \sigma_1,...,\sigma_{i-1})$, and CP with Perplexity, which applies length normalization and commonly used in language modeling, $perplexity(y) = \exp\{-\frac{1}{k}\sum_{i=1}^{k} \log p(\sigma_i | \sigma_1,...,\sigma_{i-1})\}$. Table A1 shows that for all three settings, using either sentence-level score leads to worse performance, and performance degradation correlates with variance of the multiple choice lengths. We also notice that there is no significant difference in performance between CP with RawProb and CP with Perplexity; this could indicate that beyond eliminating length bias, applying MCQA setup in KNOWNO affords additional benefit of soliciting more accurate confidence from the LLM.

| Setting | Multiple Choice Length Variance | Method | Set Size | Help |
|---------|:---:|---|:---:|:---:|
| Attribute | 1.52 | KNOWNO | 1.18 | 0.18 |
|  |  | CP w/ RawProb | 1.33 | 0.32 |
|  |  | CP w/ Perplexity | 1.35 | 0.36 |
| Spatial | 2.81 | KNOWNO | 2.23 | 0.69 |
|  |  | CP w/ RawProb | 2.50 | 0.82 |
|  |  | CP w/ Perplexity | 2.67 | 0.88 |
| Numeric | 8.51 | KNOWNO | 2.17 | 0.79 |
|  |  | CP w/ RawProb | 4.06 | 1.00 |
|  |  | CP w/ Perplexity | 4.06 | 1.00 |

Table A1: Comparison of KNOWNO with CP with sentence output and perplexity score in the three ambiguity settings in Simulation. $\epsilon = 0.15$. See Fig. A9 for samples of multiple choices generated with varying lengths.

**Potentially stronger baselines with model fine-tuning.** In Section 4 we introduce the two prompt-based baselines Prompt Set and Binary, and demonstrate them being (1) inflexible (not allowing controlling the target success rate) and (2) do not properly model the uncertainty. We note that these two baselines can be potentially strengthened by fine-tuning the LLM to better predict the binary uncertainty or the

```
We: You are a robot, and you are asked to move objects to precise locations on the table. Our instructions
can be ambiguous.

We: On the table there are these objects: green block, yellow bowl, yellow block, blue bowl, blue block,
green bowl.
We: Now, put two blocks at the blueish bowl.
You: I will put green block and yellow block on blue bowl.

We: On the table there are these objects: yellow bowl, blue bowl, green block, yellow block, blue block,
green bowl.
We: Now, put a couple of receptacles at the right side of the blue block.
You: I will put yellow bowl and blue bowl to the right of blue block.

We: On the table there are these objects: green block, yellow bowl, yellow block, blue bowl, blue block,
green bowl.
We: Now, put one bowl at the front of amber-colored block.
You: I will put green bowl at the front of yellow block.

We: On the table there are these objects: green block, blue block, green bowl, yellow bowl, blue bowl,
yellow block.
We: Now, put all cuboids at the right side of the yellow round object.
You: I will
```

Figure A8: Prompt used when evaluating the sentence-level scores in Simulation. We first generate the multiple choices in the same way as KNOWNO, and then evaluate the score of each multiple choice with this prompt. This prompt is also used to generate the option in the Binary baseline.

uncertainty set, if the true labels can be properly defined. In fact, some recent work [24, 23] have explored model fine-tuning and exhibiting the effectiveness of Binary for uncertainty calibration. We also explored fine-tuning the GPT3 model (*davinci*) from OpenAI, which is the most powerful one from OpenAI available for fine-tuning. However, we find the model performing at very low accuracy with MCQA, and fine-tuning the model always results in overfitting to the dataset, even with thousands of data and varying hyperparameters (including ones from [24]). We suspect that our scenarios exhibit high complexity and variance, and it is non-trivial to fine-tune the model well with our dataset. Nonetheless, we do hope to have future work looking into better training the model for proper uncertainty, and then applying CP on top of it to achieve set-based calibration.

```
# Attribute ambiguity
We: On the table there are these objects: green block, blue bowl, yellow block, green bowl, yellow bowl,
blue block.
We: Now, put the yellow square object at the right side of the orange container.
You:
A) put yellow block to the right of green bowl
B) put yellow block to the right of yellow bowl
C) put yellow block to the right of blue bowl
D) an option not listed here
E) put yellow block to the right of blue block

# Numeric ambiguity
We: On the table there are these objects: yellow bowl, blue bowl, green block, yellow block, blue block,
green bowl.
We: Now, put a handful of receptacles at the right side of the blue block.
You:
A) an option not listed here
B) put yellow bowl and blue bowl to the right of blue block
C) put green block and yellow block to the right of blue block
D) put yellow bowl and blue bowl and green bowl to the right of blue block
E) put green block and yellow block and blue block to the right of blue block

# Spatial ambiguity
We: On the table there are these objects: green bowl, yellow bowl, blue bowl, yellow block, green block,
blue block.
We: Now, put the yellow block along the line of sight of the blue square object.
You:
A) an option not listed here
B) put yellow block behind blue block
C) put yellow block to the left of blue block
D) put yellow block in front of blue block
E) put yellow block to the right of blue block
```

Figure A9: Sample multiple choices generated with Attribute, Numeric, and Spatial ambiguities in Simulation.

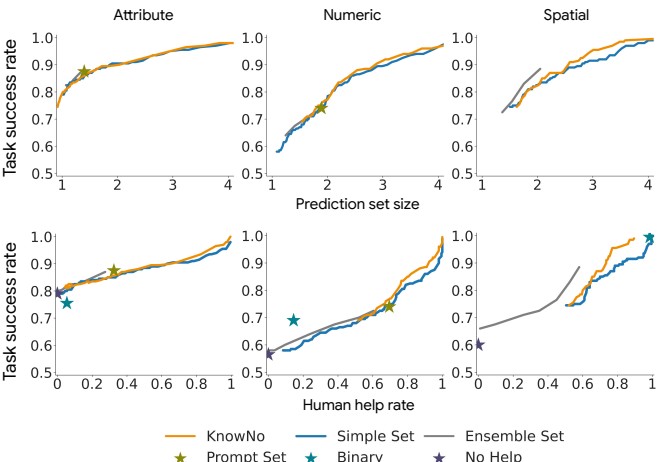

Figure A10: Comparison of task success rate vs average prediction set size (Top row) and vs. human help rate (Bottom row) in Simulation for each of the three ambiguity settings: Attribute, Numeric, and Spatial (Columns), as $\epsilon$ varies from 0.25 to 0.01 for KNOWNO, and from 0.6 to 0.01 for Simple Set and Ensemble Set. Binary and No Help are not shown in the top row since prediction set is not involved. Prompt Set does not show up in the human help rate plot for the Spatial setting as it only achieves 35% task success. As the setting gets more ambiguous (least in Attribute and most in Spatial), the performance difference between KNOWNO and Simple Set grows.

```
We: On the counter, there is a RedBull, a bottled unsweetened tea, and a bag of multigrain chips.
We: I drank it already. Can you dispose of it?
You:
A) an option not listed here
B) pick up the redbull and put it in the landfill bin
C) pick up the bottled unsweetened tea and put it in the landfill bin
D) pick up the bottled unsweetened tea and put it in the recycling bin
E) pick up the redbull and put it in the recycling bin
True label: ['D']; Prediction set with KnowNo: ['D']; Prediction set with Simple Set: ['D', 'E']

We: On the counter, there is a bottled tea, a RedBull, and a Pepsi.
We: Bring me something to drink.
You:
A) an option not listed here
B) pick up the redbull
C) pick up the pepsi
D) pick up the bottled tea
E) pick up the bottled tea and put it in the recycling bin
True label: ['B', 'C', 'D']; Prediction set with KnowNo: ['D']; Prediction set with Simple Set: ['A', 'D']

We: On the counter, there is a Sprite, a bag of kettle chips, and a Coke.
We: Put the soda in the drawer.
You:
A) open the top drawer and put the coke in it
B) an option not listed here
C) open the middle drawer and put the coke in it
D) open the middle drawer and put the sprite in it
E) open the top drawer and put the sprite in it
True label: ['A']; Prediction set with KnowNo: ['A']; Prediction set with Simple Set: ['A', 'C', 'E']

We: On the counter, there is an apple, a orange, and an energy bar.
We: Bring me a fruit.
You:
A) an option not listed here
B) pick up the apple
C) pick up the orange
D) pick up the energy bar
E) pick up the apple and put it in the landfill bin
True label: ['B', 'C']; Prediction set with KnowNo: ['B']; Prediction set with Simple Set: ['A', 'B']

We: On the counter, there is an apple, a bag of multigrain chips, and a bag of rice chips.
We: I want a healthy snack.
You:
A) pick up the rice chips
B) pick up the multigrain chips
C) pick up the apple
D) pick up the apple and put it in the compost bin
E) an option not listed here
True label: ['B', 'C']; Prediction set with KnowNo: ['C']; Prediction set with Simple Set: ['A', 'C']

We: On the counter, there is an orange soda, a Sprite, and a Pepsi.
We: Can you put away the soda? It should have expired.
You:
A) pick up the pepsi and put it in the recycling bin
B) an option not listed here
C) pick up the sprite and put it in the recycling bin
D) pick up the orange soda and put it in the compost bin
E) pick up the orange soda and put it in the recycling bin
True label: ['E']; Prediction set with KnowNo: ['E']; Prediction set with Simple Set: ['D', 'E']

We: On the counter, there is a bag of jalapeno chips, a bag of multigrain chips, and a bag of rice chips.
We: Bring me a bag of chips.
You:
A) pick up the multigrain chips
B) pick up the rice chips
C) pick up the jalapeno chips
D) an option not listed here
E) pick up the jalapeno chips and put it in the recycling bin
True label: ['A', 'B', 'C']; Prediction set with KnowNo: ['A']; Prediction set with Simple Set: ['A', 'E']
```

Figure A11: Randomly sampled scenarios from Mobile Manipulation, where KNOWNO generates singleton prediction sets that contains one of the true labels and avoids human intervention, while Simple Set generates non-singleton prediction sets and asks for human help. In Table 2 we show that KNOWNO reduces the human intervention rate by 14%. We also point out the third example where the instruction is to 'Put the soda in the drawer', which is ambiguous in the choice of top and middle drawer. In this scenario, it happens that the human means the top drawer, and KNOWNO generates the prediction set that only include the option 'open the top drawer and put the coke in it'. This example exhibits the inherent bias in LLM (e.g., bias towards top drawer over middle drawer).

