# OpenReview forum: "Robots That Ask For Help: Uncertainty Alignment for Large Language Model Planners"
_robot-learning.org/CoRL/2023/Conference — CoRL 2023 Oral_

### Official Review · Reviewer_mihK · 2023-07-18

**Confidence:** 5
**Originality:** Very Good
**Technical Quality:** Very Good
**Clarity Of Presentation:** Very Good
**Impact:** 3

**Recommendation:**

Strong Accept: I recommend accepting the paper and will argue for my recommendation even if other reviewers hold a different opinion.

**Review:**

The paper addresses the challenging problem of dealing with "confidently hallucinated outputs" that occurs when leveraging LLMs to create robot plans.
The framework makes use of the recent advantages of applying the concept of conformal predictions to LLM outputs.
The idea is straightforward and clearly motivated.
The paper is generally well written, and readers can follow the proposed framework well.
The figures explaining the framework are helpful to understand various aspects of KnowNo.
It’s great to see that the framework has also been applied to real hardware.
The submitted videos nicely visualize and summarize the conducted experiments.


Although the paper proposes an interesting framework using conformal prediction, the approach's technical description, experimental validation, and discussion open more questions than providing answers.
Conformal prediction is a powerful tool, but its coverage depends on the right calibration set.
At the moment, the paper does not describe how to create such calibration datasets.
Especially for language planning, creating calibration datasets is a tedious and time-consuming task.
For sequence plans, this process is even more tedious.
How can designers make sure that they created a good set?
Moreover, the ambiguity of language makes it difficult to find the "true" label.
The assumption that there is a "unique correct candidate plan" is unrealistic for many language-based planning tasks and not applicable when one wants to use language-based planning.
This crucial part of the approach requires much more attention in the paper.
As indicated in the paper, coverage would require new fresh calibration datasets.
To alleviate this, the paper leverages a dataset-conditional guarantee (see Eq. 2) but does not go into detail about it.
This part requires more explanation and clarifications why it hold for the considered LLM-based planning application.

The paper nicely motivates the task success coverage by leveraging human inputs whenever there is ambiguity.
This idea is present in the introduction and in a short paragraph later.
Yet, the approach and the experiments itself do not really demonstrate and analyze how well the human help works.
For simple tasks (like the placing of plastic or metal bowl), humans might be able to help.
But in other instances, such as when considering preferences, the human help might result in problems later during sequenced plans.
For instance, due to a mismatch of what the human thinks the robot can do.
It would be great if the paper analyzes how well the human help concept works and across a randomized set of tasks.
Also, the concept of preferences is introduced throughout the paper but not really the focus of the original idea.
It disturbs the reading flow and is also not well evaluated.
And, as described in the paper's discussion, preferences are not well covered yet.
If the paper wants to include preferences, it should also be validated against, e.g., preference-learning baselines.
Otherwise, the paper can also remove the "focus" on humans and remove those paragraphs.
This would also save space to describe more important technical details.
To this end, the paper sometimes reads as sections have been written with different styles and levels of side notes.
In my opinion, many side notes (e.g., "multiple acceptable options") are thrown into the context without proper explanation and motivation.
To simplify the paper's flow, it would be better for readers if the paper focuses on the core idea.
The gained space can also help again to improve important details.
Moreover, LLM-based planning will (in most situations) make use of sequenced planning (time series).
So why not directly describe this setting and not spend a lot of time on the single planning case and reiterating and abusing notations to introduce the sequenced version?


The experiments have been carried on multiple platforms and tasks, and mostly look good.
Yet, to my understanding, the experimental validation is often handpicked, and the considered scenarios are overly simple.
For instance, Exp. 1 considers scenarios with just one alternative (choose between two clear objects) and Exp. 2 focuses on preferences placing items in one of two possible goal locations with single planning steps instead of sequences.
Are those the difficult hallucinated plan options that the introduction mentions?
I wonder if these are the realistic problem cases where the approach is needed.
The experiments would better show the approach's benefits in situations where the robot has much more options to choose from or in situations where the LLM hallucinates actions and plan sequences.
An example could be to task the robot to set the dinner table or to clean up the office space.
In the experiment section, there is also no clear description of details, such as how the calibration datasets have been created or which scoring functions have been used.
This makes it difficult to reproduce the results of the paper.
The description and discussion of the results is short (probably giving the space constraints), and it is often difficult to follow the claims.
The provided numbers or briefly mention but not discussed well to draw proper conclusions, e.g., in Fig. 3: how many examples have been used?  and in Fig. 4: where is the binary label in the left figure or why are there continuous graphs for a discrete prediction set size?
The analysis of using other LLMs draws no conclusion.
The considered baselines are not well described, and it is not clear how they are exactly implemented and why they have been chosen.
For instance, the ensemble set seems to work well, but it is later in L253 discarded without providing enough details.
The experiments later (4.3) on compared to the simple set approach.
Why does the experiments now focus solely on this baseline?
The claim of using human help only when needed is evaluated in one single experiments.
I wonder why this is not reported for all conducted experiments, since this is a core claim of the paper?

Lastly, the limitation section is another weak part of the paper.
It is not a sophisticated discussion of the approach but rather a summary of future work.
This approach makes many assumptions that are scattered across the paper, e.g., "unique correct candidate plans".
These are strong assumptions and need to be discussed.
Similarly, the creation of the calibration dataset is what provides the coverage guarantees.
There is no discussion of the calibration dataset and how difficult the creation of such datasets is.
The limitations section requires a major revision to truly discuss the current shortcomings of KnowNo when it can be applied and when it fails.


Other minor things are:
- The paper introduces POMDPs but does not use the concept throughout the paper. Why are POMDP introduced and what is partially observable?
- The variable C is overloaded and used as the prediction set and the answer in the MCQA.
- The variable s is used to denote the state and the score
- Some notations are not introduced, e.g., Eq. 4 t\in[T] (not even a proper interval definition) or \hat{f}(\tilde{x}^t))_y?
- The package cite would help to avoid elongated citation number lists

**Quality Of The Limitations Section:**

Limitations are addressed clearly

**Questions For Rebuttal:**

* What happens in situations where there is only one option? How does the multiple-choice answers work here?
* What is the conclusion from the conducted experiments with various LLMs?
* How are the calibration datasets generated? What assumptions are made and how can one determine the true labels?
* How realistic is the assumption of having a "unique correct candidate plan"?
* How does human help perform across all experiments?
* What are the limitations of the approach?
* Which assumptions does the approach make?
* How does the approach perform on less simplistic scenarios and tasks?
* How are preferences covered in the approach and in the calibration dataset?
* How does the dataset-conditional guarantee work?

**Robotics Focus:**

Sufficient demonstration on hardware

**Summary Of Paper:**

This paper presents the framework KnowNo to provide statistical guarantees on task completion of a robot instructed by a large language model (LLM).
To this end, the paper incorporates the concept of conformal predictions into an LLM-based planning framework.
Whenever the created set of predicted plans (with given task success level) contains more than one option, a human can be queried to resolve ambiguity.
The approach has been evaluated on multiple tasks in simulation and on a real robot.

**Summary Of Recommendation:**


The paper presents an interesting addition to LLM-based planning for robots by leveraging the powerful concept of conformal predictions.
The motivation is clear, and the authors provide a sophisticated framework with a great potential.
Yet, the approach is not well described and discussed.
Details that are crucial for the approach's robustness, such as how the calibration datasets are created, are not described in enough detail, and discussed.
The experimental validation, although on multiple tasks, is too simplistic to validate the advantages and claims of the paper.
The approach needs to be validated across scenarios with more objects and ambiguity, or on randomized tasks.
Lastly, the limitations section needs a major revision to properly discuss the approach's assumptions, how to set it up, when it works and when doesn't.

---

### Official Review · Reviewer_aKea · 2023-07-20

**Confidence:** 4
**Originality:** Good
**Technical Quality:** Very Good
**Clarity Of Presentation:** Very Good
**Impact:** 4

**Recommendation:**

Strong Accept: I recommend accepting the paper and will argue for my recommendation even if other reviewers hold a different opinion.

**Review:**

Strengths:
1. The paper is well written, and the presentation is clear enough for the theoretical part.
2. To the best of our knowledge, using conformal prediction to estimate uncertainty to improve LLM-based planners for robotics is novel.
3. Sufficient experiments in the real world.

Weaknesses:
1. As mentioned in the limitation subsection, the method assumes a perfect textual description of the scene. The method considers only epistemic uncertainty while neglecting epistemic uncertainty coming from data. It would be interesting to discuss ways to estimate the observation's uncertainty and how to handle it.


**Quality Of The Limitations Section:**

Limitations are addressed clearly

**Questions For Rebuttal:**

No major questions

**Robotics Focus:**

Sufficient demonstration on hardware

**Summary Of Paper:**

The paper builds on the theory of conformal prediction to measure and align the uncertainty of LLM planners, asking for help when needed. The proposed method provides theoretical guarantees on task completion while minimizing required help. The method does not require fine-tuning the large language model. Instead, it generates multiple-choice questions and uses the estimated uncertainty to choose the suitable action.
The paper reports evaluation in situations and the real world, demonstrating the method's efficacy.

**Summary Of Recommendation:**

We recommend accepting the paper for publication. The proposed method is novel, and the results in simulation and real-world are convincing.

---

### Official Review · Reviewer_4oDG · 2023-07-20

**Confidence:** 4
**Originality:** Very Good
**Technical Quality:** Very Good
**Clarity Of Presentation:** Good
**Impact:** 3

**Recommendation:**

Weak Accept: I recommend accepting the paper, but will not argue for my recommendation if the majority of other reviewers have a different opinion.

**Review:**

An important issue has been correctly identified in this research. The reviewer appreciates that this paper aims to address a fundamental challenge in LLM-based planning systems. The CP approach, as a statistical technique for computing confidence levels, is a well selected approach to address this issue.

If the reviewer understands it correctly, the single-step setting considers the tasks that require only one action (Section 3.2). If that's the case, are those tasks still pointing to "planning" problems? Some clarification would be appreciated.

How was the iid scenario dataset (with 400 instances) collected? It seems to be very important because the effectiveness of CP highly relies on the assumption that the labeled and unlabeled data instances are "exchangeable". Since the calibration dataset was collected before the execution time, how is the exchangeability guaranteed? The reviewer looks for a clear response to this question.

Continuing the above comment, there's the statement that "... to ensure the desired probability of coverage for each new ztest, one needs a fresh calibration set. But, in practice, we only calibrate once with a fixed set."  The reviewer is not sure how to interpret this statement. Does it mean that the whole algorithm loses the theoretical foundation in this research? Some elaborations about the gap between the fixed calibration set and the desired fresh set would be helpful.

Another concern is that in this research, human models are greatly simplified, which negatively affects the applicability of the developed approach in the real world. For instance, the human is assumed to be always available and equipped with complete knowledge, and no difference is assumed among different types of help from people (cost, accuracy, etc). While it's not the focus of this research, some discussions about human modeling would be nice.

There should be more advanced ways of using the calibrated confidence level of task plans. The current way of using it is quite simple -- if there are multiple plans above a quality threshold, the robot asks human to make the selection. Many other factors could be considered (including those about humans in the last comment) for better leveraging the uncertainty analysis -- some discussions about that would be nice.

**Quality Of The Limitations Section:**

Additional details required

**Questions For Rebuttal:**

See above.

**Robotics Focus:**

Sufficient demonstration on hardware

**Summary Of Paper:**

LLMs have been used for robot planning in recent literature. This work is motivated by the "hallucination" issue of LLMs that tend to generate plans that are plausible in general but are incorrect or infeasible in practice. Looking into such issues, it's found that the current LLM-based planning systems lack the capability of leveraging its own confidence level in selecting a plan solution among those that are potentially feasible. This paper introduces a new algorithm and system called KnowNo that uses Conformal Prediction (CP) to align the uncertainty of LLM plans. More specifically, the LLMs generate a set of possible next-step actions, and ask help from human for decision making only if there are more than one above-CP-quantile solutions. KnowNo has been evaluated both in simulation and using a real robot in completing object rearrangement tasks.

**Summary Of Recommendation:**

This research is timely and addresses an important problem in LLM-based robot planning. The developed approach makes some approximations that can potentially break the theoretical guarantees, where some clarification are needed.

---

### Official Review · Reviewer_fuC2 · 2023-07-23

**Confidence:** 5
**Originality:** Good
**Technical Quality:** Good
**Clarity Of Presentation:** Good
**Impact:** 3

**Recommendation:**

Weak Accept: I recommend accepting the paper, but will not argue for my recommendation if the majority of other reviewers have a different opinion.

**Review:**

The basic idea proposed is intuitive and appears to generally benefit system performance (aka successful task completion).

In addition to the questions below, my main concern with this paper is not being properly situated in the literature.  There is a long history of people working on robot dialogue both within simulation and on physical hardware, for navigation and manipulation.  I don't feel it's appropriate for me to do a full literature review here but CVDN, TEACh, Dialfred are recent in simulation, RMM, Jarvis are recent models, RobotSlang from CoRL 2020, and SigDial has run entire special tracks on robot dialogue etc, but none of this literature is cited here.  I think the authors can claim there are important differences about how their proposal is tied to calibration on LLMs, but there's also a large literature on language generation evaluations that's not cited here -- I will note there is more awareness of the LM literature in the related work, but not a direct discussion of how what's proposed here and the claims connect to standard practice, whether it's BARTScore, T5Score, etc which have lots to say about the issue of length penalties and so forth (both within semantic similarity monolingually and machine translation).  There are just a lot of places where the paper reads like the authors are the first people to look at these issues. The last literature missing is that of diverse generations, specifically diverse captioning, which a cursory read through CV proceedings will dig up a large literature -- likely very beneficial to the Ensemble Set definition.

**Quality Of The Limitations Section:**

Limitations are addressed clearly

**Questions For Rebuttal:**

This brings me to my questions:

1. How would a more diverse captioning (or sampling - e.g. https://arxiv.org/abs/1904.09751) change the results for the Simple/EnsembleSet and would that close the gap with the KnowNo approach?
2. It is unclear to me how significant the differences in Table 1 and Figure 4 are?  How many examples separate these approaches?
3. [nice to have] How is performance impacted by the scene/task complexity? Is part of the reason the approach works that it's easy to get good coverage over the space of next actions/objects?

**Robotics Focus:**

Sufficient demonstration on hardware

**Summary Of Paper:**

The paper looks at enabling robots to compare possible next steps in a task plan and use their calibrated uncertainty to indicate when to ask a question -- for example, to clarify some ambiguity in the scene. This is an important ability for our agents to have -- both for their own success and for alignment to human preferences.

**Summary Of Recommendation:**

I think this is a valuable direction for the community to be moving, and I think the proposals are reasonable for the scope of a paper -- but a few questions are left open and the work does not articulate how its claims/approaches are related to or differ from what is already in the literature.

---

### Author Response · Authors · 2023-08-09
**General response from the authors**

We thank the reviewers for the detailed feedback. We are happy to see that the reviewers appreciated (1) the strong motivation of our work in addressing LLM hallucinations in the context of robot planning, (2) the intuitive and well-fitted application of conformal prediction, and (3) sufficient demonstration of the results on hardware. Among the concerns raised, we agree that presentation of the work can be improved and we have revised the paper accordingly, especially including more details on the scenario distributions in different experiments and how the calibration datasets are generated. We have also included more discussions on the experimental results based on the reviewers’ feedback. Please see the revised manuscript attached in each individual rebuttal, which also includes the revised appendix.

---

### Decision · Program_Chairs · 2023-08-30

**Decision:**

Accept (Oral)

**Comment:**

The paper combines large language model (LLM) planning with uncertainty calibration to enable a robot to ask clarifying questions of humans. The paper builds on incorporating conformal predictions into the LLM planning and provides statistical guarantees on task completion. When it is unclear which option to follow, the robot is able to ask a human to clarify. For example, picking up a drink could mean picking up a coke or lemonade. A human can clarify this. Overall, the paper is of high quality with results in both simulation and on a real robot.